# Changes in Etiology and Clinical Outcomes of Pleural empyema during the COVID-19 Pandemic

**DOI:** 10.3390/microorganisms11020303

**Published:** 2023-01-24

**Authors:** King-Pui Florence Chan, Ting-Fung Ma, Siddharth Sridhar, David Chi-Leung Lam, Mary Sau-Man Ip, Pak-Leung Ho

**Affiliations:** 1Department of Medicine, University of Hong Kong, Queen Mary Hospital, Hong Kong SAR, China; 2Department of Statistics, University of South Carolina, Columbia, SC 29208, USA; 3Department of Microbiology, Carol Yu Centre for Infection, University of Hong Kong, Hong Kong SAR, China

**Keywords:** pleural empyema, COVID-19, etiology, health-seeking behavior

## Abstract

Healthcare-seeking behavior changed during the COVID-19 pandemic and might alter the epidemiology of pleural empyema. In this study, the incidence, etiology and outcomes of patients admitted for pleural empyema in Hong Kong in the pre-COVID-19 (January 2015–December 2019) and post-COVID-19 (January 2020–June 2022) periods were compared. Overall, *Streptococcus pneumoniae* was the predominant organism in <18-year-old patients, while *Streptococcus anginosus,* anaerobes and polymicrobial infections were more frequent in adults. In the post-COVID-19 period, a marked decline in the incidence of pleural empyema in children was observed (pre-COVID-19, 18.4 ± 4.8 vs. post-COVID-19, 2.0 ± 2.9 cases per year, *p* = 0.036), while the incidence in adults remained similar (pre-COVID-19, 189.0 ± 17.2 vs. post-COVID-19, 198.4 ± 5.0 cases per year; *p* = 0.23). In the post-COVID-19 period, polymicrobial etiology increased (OR 11.37, *p* < 0.0001), while *S. pneumoniae* etiology decreased (OR 0.073, *p* < 0.001). In multivariate analysis, clinical outcomes (length of stay, ICU admission, use of intrapleural fibrinolytic therapy, surgical intervention, death) were not significantly different in pre- and post-COVID-19 periods. In conclusion, an increase in polymicrobial pleural empyema was observed during the pandemic. We postulate that this is related to the delayed presentation of pneumonia to hospitals.

## 1. Introduction

Pleural infection, especially complicated parapneumonic infection including pleural empyema, is a clinical problem associated with high mortality and morbidity. Drainage is frequently required, and prolonged courses of antibiotics are needed [1]. The yield of positive pleural fluid culture was reported to be low [2]. Community-acquired pleural empyema can be either monomicrobial or polymicrobial. Common microbial etiologies include *Streptococcus* species (including *Streptococcus pneumoniae* and *Streptococcus anginosus* group), *Staphylococcus aureus*, and oral anaerobes (including *Fusobacterium* species) [3]. *Mycobacterium tuberculosis* (MTB) is an important cause of pleural empyema in endemic areas [4]. Treatment of pleural empyema requires antibiotics tailored to culture results [1]. Intrapleural fibrinolytic therapy (IPFT) in pleural empyema can also be considered for those failed initial drainage aims to decrease rates of surgical referral and hospital length of stay [5]. Surgical drainage and decortication might be needed if there are suboptimal clinical responses or uncontrolled infection. In retrospective series, the mortality of pleural empyema was approximately 15% and was higher in hospital-acquired infections, elderly and in the presence of multiple comorbidities [3,4]

Coronavirus disease 2019 (COVID-19) caused by Severe Acute Respiratory Syndrome Coronavirus 2 (SARS-CoV-2) infection was declared as a pandemic by the World Health Organization (WHO) on 11 March 2020 [6]. In response to the pandemic, universal masking and social distancing were widely practiced in different places around the world, including Hong Kong [7,8]. Consequently, there have been reports of decrease in incidence of influenza, invasive *S. pneumoniae* and *Haemophilus influenzae* infections [7,8,9,10]. 

On the other hand, changes in healthcare-seeking behavior were also reported, with a decrease in emergency department visits during early COVID-19 periods [11] and increase in mortality of non-COVID illnesses [12] due to possible fear of being infected by COVID-19 [13]. Delayed presentation to medical care has been observed in several countries for both acute and chronic diseases across different age groups. The report in delayed presentation to emergency departments was as high as 52% in Saudi Arabia and Bahrain, leading to a significantly higher mortality rate in these patients [14]. Delayed presentation was also reported in emergency conditions including acute ischemic stroke in the United States [15] and myocardial infraction in Hong Kong, with increase in door-to-device time [16].

The impact of altered respiratory microbial etiology and healthcare-seeking behavior during the COVID-19 pandemic on pleural empyema incidence and patient outcomes is unclear. The objective of this study was to compare the incidence, microbiology and patient outcomes of pleural empyema during the COVID-19 pandemic to previous periods.

## 2. Materials and Methods

### 2.1. Study Design

This was a territory-wide retrospective study on patients admitted to public hospitals under the Hospital Authority for pleural infection with positive pleural culture results between January 2015 and June 2022. January 2020 to June 2022 was defined as the post-COVID-19 period, and January 2015 to December 2019 was defined as the pre-COVID-19 period. The Hospital Authority is a public organization which provides 90% of in-patient services in Hong Kong [17], serving the city’s 7.5 million population. Public hospitals under Hospital Authority provide care to patients with both COVID-19 infection and other acute admissions through emergency departments. 

### 2.2. Data Source

Data were collected from a territory-wide healthcare database of Hospital Authority, which is known as Clinical Data Analysis and Reporting System (CDARS). Patients with International Classification of Diseases, ninth revision (ICD-9) code 510 (pleural empyema) with in-patient stay were searched. 

### 2.3. Inclusion/ Exclusion Criteria

Patients with pleural empyema and definite/ probable microbial etiology identified were included. Definite microbiological etiology included any one of the following: (i) pleural fluid culture positive, (ii) pleural fluid with *Mycobacterium tuberculosis* DNA detected by polymerase chain reaction (PCR), and (iii) pleural fluid with pneumococcal DNA detected by PCR. Probable microbiological etiology included the following: (i) sputum or tracheal aspirate culture positive for *Mycobacterium tuberculosis* or MTB-PCR; (ii) blood culture positive for *Streptococcus pneumoniae*; and (iii) pneumococcal urine antigen test (UAT) positive. Patients with ICD-9 code 510 but with negative culture results or microbiological workup not fitting the above criteria were excluded. 

### 2.4. Outcome

The primary outcome was the microbial etiology of pleural empyema. Secondary outcomes included hospital length of stay (LOS), ICU admissions, need for IPFT and surgical intervention. In our locality, IPFT consisted of the combined use of tissue plasminogen activator (tPA) and recombinant deoxyribonuclease (DNase). Surgical procedures included decortication of lung or lobectomy.

### 2.5. Statistical Analysis

Analysis and comparison were performed on the data between the pre and post-COVID-19 period by the International Business Machines Corporation Statistics Package for the Social Sciences (IBM SPSS) statistics version 27. Patient demographics and outcome between the two periods were compared by Fisher’s exact test or Wilcoxon rank-sum test [18]. Differences in the microbiological results including percentage of polymicrobial, anaerobes between different periods were investigated by Fisher’s exact test. Further analysis was performed by first calculating the odd ratios together with the 95% confidence interval (95%CI) using SPSS. Odd ratios were calculated based on the proportion. Patient outcomes in the two periods, including mortality, need of surgical treatment, need of IPFT and length of hospital stay, were first compared by Fisher’s exact test for any difference. It would then be followed by multivariable analysis through linear regression [19] for interval-dependent variable, namely length of stay, or by logistic regression [20] on the possible effect of different factors, namely age, sex, and Charlson’s comorbidity index (CCI) score. The values of parameters are given as mean (± standard deviation [SD]) or median (± interquartile range (IQR)) where appropriate. All reported *p* values were two-sided. A *p* value of less than 0.05 was considered to be statistically significant.

## 3. Results

### 3.1. Patients Demographics

In total, 1513 pleural empyema cases were included (Figure 1), with 1014 cases in pre- and 499 in the post-COVID-19 period. Overall, 75.3% were male and 24.7% were female patients. The proportions of male patients in the pre- and post-COVID periods were similar (74.9% vs. 76.4%, respectively, *p* = 0.52). Among the cases, 58 (3.8%), 39 (2.6%), 465 (30.7%), 394 (26.0%) and 557 (36.8%) were patients aged ≤5 years, 6–17 years, 18–59 years, 60–69 years and ≥70 years, respectively. For adult cases, the median age was 65 ± 15.4 and 67 ± 15.2 in the pre and post-COVID-19 period, respectively (*p* = 0.14). The median CCI score of cases in the pre- and post-COVID-19 period was 3 ± 3.0 and 4 ± 2.8, respectively (*p* < 0.001).

### 3.2. Pleural Empyema Etiology

Among the 1513 pleural empyema cases, 87.0% and 13.0% were of monomicrobial and polymicrobial etiology, respectively. In 1413 (93.4% of total) cases, positive pleural cultures or positive pleural PCR results were obtained, including 1217 positive for one organism and 196 positive for two or more organisms. In the remaining 100 culture-negative cases, 86 (5.7% of total) and 14 (0.9% of total) cases, respectively, were considered to be probable pneumococcal and probable MTB, respectively, by pneumococcal UAT and positive MTB-PCR in non-pleural respiratory specimens. The infecting organisms were classified into 14 groups (Table 1). Anaerobes, *S. anginosus* group and *Staphylococcus aureus* were detected most frequently in both monomicrobial and polymicrobial infections. The organisms in polymicrobial infections were diverse. Species of oropharyngeal flora including *Bacteroides*, *Fusobacterium*, *Parvimonas* and *Peptostreptococcus* were the most frequent anaerobic bacteria. Anaerobic bacteria were detected in 19.7% of monomicrobial and 56.6% of polymicrobial infections. The *S. anginosus* group was detected in 21.9% of monomicrobial and 26.5% of polymicrobial infections. *S. aureus* was detected in 12.4% of monomicrobial and 12.8% of polymicrobial infections. Among the 188 cases positive for *S. aureus*, 47% were positive for methicillin-resistant *S. aureus* (MRSA). *Streptococcus pneumoniae* was present in 9.3% of cases. *Klebsiella pneumoniae* and other *Enterobacterales* were present in 5.9% and 7.9%, respectively.

### 3.3. Incidence of Pleural Empyema and Etiology by Time Periods and Age Groups

The mean ± number of pleural empyema in patients of all ages in the pre-COVID-19 and post-COVID-19 periods were 203 ± 21.9 cases per year and 197 ± 5.0 cases per year, respectively (*p* = 1). In the post-COVID-19 period, a marked decline in the number of pleural empyema in children was observed. In children aged 0–17 years, the average number of pleural empyema decreased from 18.4 ± 4.8 cases per year in the pre-COVID-19 period to 2.0 ± 2.9 cases per year in the post-COVID-19 period (*p* = 0.036). In adults (aged ≥18 years), the average number of pleural empyema in the pre- and post-COVID-19 periods were 189.0 ± 17.2 cases per year and 198.4 ± 5.0 cases per year, respectively (*p* = 0.23). Among patients aged ≤5 years and 6–17 years, *S. pneumoniae* was the predominating organism identified (Figure 2). By comparison, *S. anginosus* group, anaerobes and polymicrobial infections were most frequent in adults. An abrupt increase in the proportion of polymicrobial pleural empyema was observed in the post-COVID-19 years. The annual proportion of polymicrobial infection during the pre-COVID-19 period ranged 1.6%-4.9%. This increased to 30.7%-40.6% in the post-COVID-19 period (*p* < 0.001). All polymicrobial infections involved adults. The proportion of polymicrobial etiology among patients aged 18–59 years, 60–69 years and ≥70 years were 3.8%, 2.8% and 2.8%, respectively, in the pre-COVID-19 period, and 40.8%, 31.0% and 30.5%, respectively, in the post-COVID-19 period. The odds ratio of polymicrobial etiology was 11.37 (95% CI 7.6–17.1, *p* < 0.0001) in the post-COVID-19 period compared to the pre-COVID-19 period (Figure 3). There was an abrupt decrease in *S. pneumoniae* in the post-COVID-19 period with an odds ratio of 0.073 (95% CI 0.030–0.181, *p* < 0.001) compared to the pre-COVID-19 period.

### 3.4. Pleural Empyema in COVID-19 Patients 

In the post-COVID-19 period, the proportion of pleural empyema with a diagnosis of COVID-19 in the same episode was 4.2% (21/499). The 21 cases included three cases in 2020, two cases in 2021 and 16 cases in 2022 (Table 2). Based on the first positive SARS-CoV-2 RNA test result, two cases were diagnosed to have COVID-19 on the day pleural empyema was diagnosed, 10 cases were diagnosed to have COVID-19 prior to pleural empyema diagnosis by a median of 29 ± 30.6 days, and nine cases were diagnosed to have COVID-19 after pleural empyema by a median of 21 ± 26.6 days. The majority (81.0%, 17/21) of them were aged ≥60 years, with median age 68 ± 20.1 years. In the only pediatric case, pleural fluid was culture positive for *S. pneumoniae*. In adult cases, the major etiologies were polymicrobial (six cases of which four cases involved anaerobes), *S. anginosus* group (five cases) and *S. aureus* (in three monomicrobial cases and two polymicrobial cases). There was no statistically significant difference in the percentage of polymicrobial pleural empyema between COVID-19-related and non-COVID-19 related cases in the post-COVID-19 period (28.6% vs. 33.7%, respectively, *p* = 0.81). The mortality rate in COVID-19 related pleural empyema was 23.8% (5/21). The odds ratio of mortality compared to patients without COVID-19 in 2020–2022 was 1.2 (95%CI 0.4–3.4) and was statistically insignificant (*p* = 0.71). One patient required surgical intervention and was statistically insignificant compared to other patients in the post-COVID-19 period (*p* = 0.499).

### 3.5. Clinical Outcomes of Pleural Empyema in the Pre- and Post-COVID-19 Periods

The median hospital LOS of patients in the two periods was not significantly different (Table 3), 24 ± 25 days in the pre-COVID-19 period vs. 23 ± 21 days in the post-COVID-19 period (*p* = 0.16). In the post-COVID-19 period, more patients with pleural empyemas were treated with IPFT than in the pre-COVID-19 period (8.4% vs. 2.5%, *p* < 0.001). Of the 1513 patients, 183 (12.1%) patients were admitted to the ICU, 69 (4.6%) patients were treated with IPFT, and 145 (9.6%) patients required surgical intervention for the pleural empyema. Overall, 291 (19.2%) patients died during hospitalization. The mortality rate in the 0–17 years age group was 1.0%, that in the 18–59 years age group was 8.8%, that in the 60–69 years age group was 18.4% and that in the ≥70 years age group was 31.5% (*p* < 0.001). Clinical outcomes including proportions of ICU admission, surgical intervention and episode death were not significantly different between the two periods (Table 3).

Results from multivariate analysis of risk factors for adverse or better clinical outcome are summarized in Table 4. Older age was significantly associated with increased risk of death (OR 1.03 for every 1 year increase in age). A higher CCI score was significantly associated with lower likelihood for surgical intervention (OR 0.78 for each unit of CCI score increment) and higher risk of death (OR 1.16 for each unit of CCI score increment). Hospital LOS was significantly longer in patients with polymicrobial etiology (10.0 days, 95% CI 4.0–16.1), while the presence of anaerobes (-13.1 days, 95% CI −7.9 to −18.2) and *S. anginosus* (-15.2 days, 95% CI −10.0 to −20.2) was associated with shorter LOS. In addition, better outcomes were associated with the presence of anaerobes (lower OR for IPFT and risk of death) and *S. anginosus* (lower OR for IPFT, surgical intervention and risk of death) (Table 4). 

## 4. Discussion

This is a territory-wide study on the microbiological etiology of pleural empyema, involving data from 1513 patients. It provides an update on the microbiology of pleural infection, which would be useful in guiding the use of empirical therapy for pleural empyema. The etiologies before and after COVID-19 emergence were also compared. Possible correlations of bacteriology and patient outcomes were found. 

We observed an overall non-significant decrease in the average annual incidence of pleural empyema in the post-COVID-19 period, which was driven by decrease in incidence in the 0 to 17-year-old patients. *Streptococcus pneumoniae* has historically been the commonest organism among this group. This echoed with our previous study which observed a decrease in incidence of pneumococcal pneumonia and invasive pneumococcal disease during the COVID-19 pandemic in Hong Kong [9]. Prior to the COVID-19 pandemic, pneumococcal serotype 3 predominated among pediatric invasive pneumococcal disease in Hong Kong, and it was often complicated by pleural empyema [21]. We postulate that non-pharmaceutical interventions such as universal masking and school closures contributed considerably to the decline in invasive pneumococcal disease during the pandemic era. 

A higher percentage of polymicrobial pleural empyema in post-COVID-19 period was observed in the current study. Species of oropharyngeal flora, of which anerobic bacteria was the commonest, were detected in more than half of the polymicrobial cases followed by *S. anginosus* and *S. aureus*. The CCI score was also higher among patients in the post-COVID-19 period than the pre-COVID-19 period and is statistically significant. The effect of healthcare-seeking behavior in delaying presentation to hospitals for patients with chronic diseases was reported in previous studies [13]. Access to dental services was limited due to lockdown, and a decrease in attendance for preventive dental care was reported [22,23]. Infections with anaerobes are more likely to have insidious clinical onset and are more common following possible aspiration pneumonia and poor dental hygiene, together with more common polymicrobial infection in patients with chronic disease [24]. Patients’ characteristics including higher CCI score, possible poorer dental hygiene due to decrease in access to dental service, as well as delaying hospital presentation, which might result in an increase in the incidence of pleural empyema with polymicrobial infection. Polymicrobial pleural empyema was associated with longer length of stay and higher mortality, although it was statistically insignificant in this cohort and was shown in another study [25]. 

Higher mortality in patients with *S. aureus*, advance age and higher CCI score were observed in this and other studies [26,27]. However, we did not see any statistically significant difference in mortality between the two periods. Our study highlighted the change in bacteriology between the pre- and post-COVID-19 periods, with a higher proportion of anaerobes and polymicrobial etiology. Since the COVID-19 pandemic is not yet over, this might provide guidance for the use of antibiotics in the treatment of pleural empyema [1], especially before culture results are available.

To the best of our knowledge, no previous population-based studies were carried out on COVID-19 related pleural empyema. This is the first territory-wide study that included patients with COVID-19 related pleural empyema with an electronic patient record system covering the vast majority of hospitalized cases allowing ample data collection retrospectively. Case reports on COVID-19 related pleural empyema reported yielding of *Pseudomonas aeruginosa* [28], community-acquired MRSA [29], *S. anginosus* or negative culture [30]. The presence of alveolar-pleural fistula was reported [28]. Two cases required surgical management apart from chest tube drainage [28,29]. Our study provided further understanding on the microbiology of COVID-19-related pleural empyema in a population-based level, which is useful in guiding the choice of antibiotics and patient treatment. The dysregulation of immune response in patients with COVID-19 was reported [31], which might be a contributory factor in bacterial co-infection leading to pleural empyema. In our study, the microbiology of COVID-19-related pleural empyema demonstrated no difference with the other cases in post-COVID-19 period, and both groups had a higher percentage of polymicrobial pleural empyema compared to the pre-COVID-19 period. On the other hand, our study also highlights the possible higher mortality of COVID-19-related empyema compared to other patients in the same period. During the study period, a total of 1,245,797 COVID-19 cases were confirmed in Hong Kong. This translates into an overall incidence of approximately one pleural empyema per 60,000 COVID-19 cases, indicating a low incidence of this complication after COVID-19.

Several study limitations should be acknowledged. First, this study compared the findings in an international city in the pre- and post-COVID-19 periods; hence, the observations may not be extended to other regions where the COVID-19 pandemic countermeasures and healthcare systems are different. Second, while data on demographics, comorbidities and the microbiology were obtained, the causative relationship cannot be ascertained because of the retrospective design. Moreover, our study also included pleural empyema patients with microbiology defined as probable microbiological etiology. This group includes those with microbe not directly identified in the pleural fluid but from sputum (sputum grew *M. tuberculosis* or positive by MTB-PCR) or blood (blood culture positive for *S. pneumoniae*) or urine (UAT positive). These patients did not obtain a positive pleural microbiological workup. The culture positive rate of tuberculosis pleurisy was not high and was quoted as 63% in a recent study [32]. Combining the sputum and pleural effusion results has been shown to have higher diagnostic yield at 79%. Hence, we believe it is reasonable to include patients with positive sputum culture for tuberculosis as having tuberculous pleural empyema instead of only including those with positive pleural microbiological results. Furthermore, many patients may have received antibiotics before pleural tap, so the culture-based microbial assessment is biased away from more sensitive organisms (particularly anaerobes). 

## 5. Conclusions

Our study reports a change in bacteriology in pleural empyema in the post-COVID-19 period, with an increase in polymicrobial pleural empyema compared to the pre-COVID-19 period. This information is useful in guiding antibiotics treatment, which would contribute toward better patient outcome. 

## Figures and Tables

**Figure 1 microorganisms-11-00303-f001:**
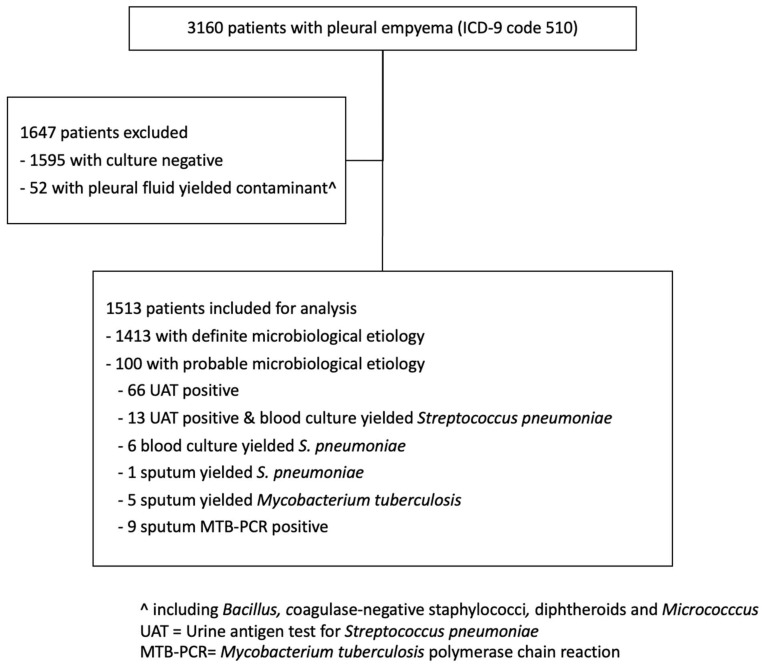
Flowchart for patients selection in the study period January 2015 to June 2022.

**Figure 2 microorganisms-11-00303-f002:**
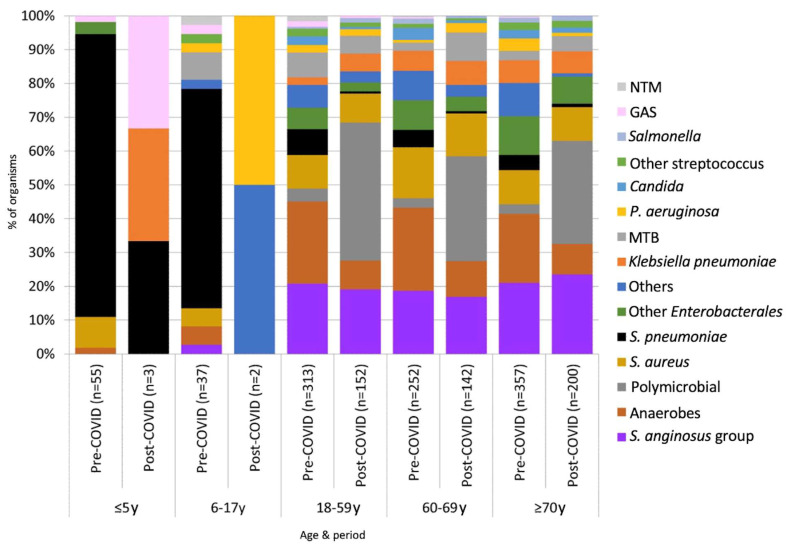
Microbial cause of pleural empyema according to age groups.

**Figure 3 microorganisms-11-00303-f003:**
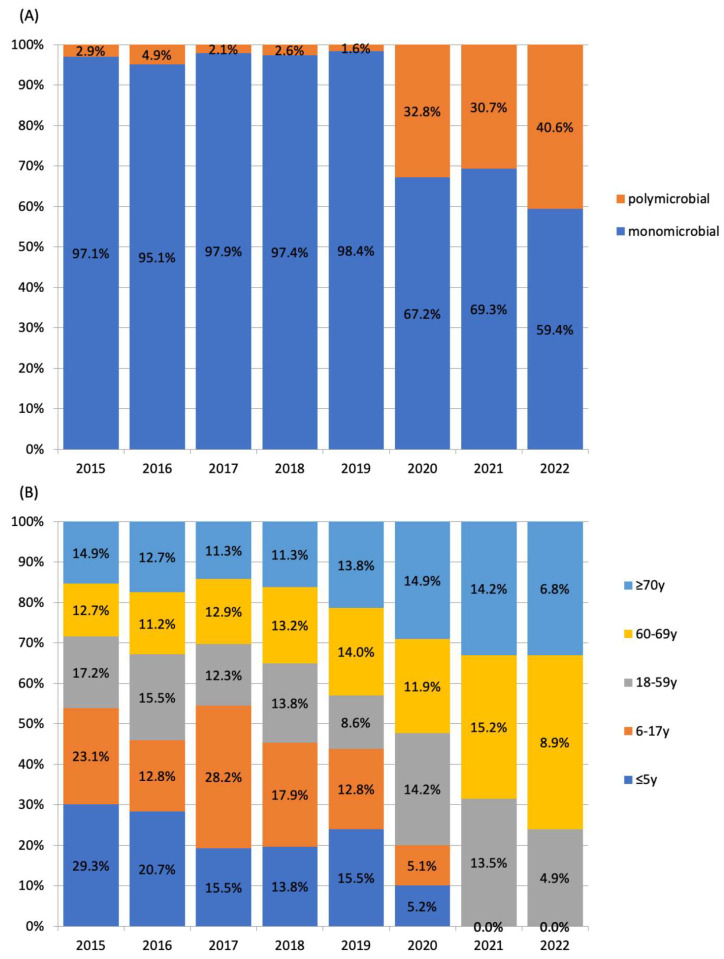
Temporal changes in (**A**) proportions of polymicrobial and monomicrobial pleural empyema and (**B**) proportions of pleural empyema by age group proportions.

**Table 1 microorganisms-11-00303-t001:** Microbiology of pleural empyema in this study.

Organism Groups	No. (%) of Pleural Empyema Containing the Organism		
	Monomicrobial	Polymicrobial a, b	Total b
Anaerobes	260	111	371 (24.5)
*Streptococcus anginosus* group	288	52	340 (22.5)
*Staphylococcus aureus*	163	25	188 (12.4)
*Streptococcus pneumoniae*	128	12	140 (9.3)
Other *Enterobacterales*	111	9	120 (7.9)
*Klebsiella pneumoniae*	78	11	89 (5.9)
MTB	71	3	74 (4.9)
*Candida*	31	15	46 (3.0)
*Pseudomonas aeruginosa*	33	9	42 (2.8)
Other streptococcus	26	4	30 (2.0)
*Salmonella*	17	1	18 (1.2)
Group A streptococcus	11	0	11 (0.7)
NTM	8	1	9 (0.6)
Others	92		92 (6.1)
Total	1317	196	1513 (100)

NTM, non-tuberculous mycobacteria; MTB, *Mycobacterium tuberculosis.* ^a^ Number of polymicrobial pleural empyema with the organism. ^b^ Since multiple organisms were be detected in the polymicrobial pleural fluid samples, the number do not add up to the total.

**Table 2 microorganisms-11-00303-t002:** Summary of 21 COVID-19 related pleural empyema cases, Hong Kong, 2020–2022.

	No. of Patients				
	≤5 y	18–59 y	60–69 y	≥70 y	Total
Etiology					
Polymicrobial ^a^		1	4	1	6
*Streptococcus anginosis*			2	3	5
*Staphylococcus aureus*			2	1	3
*Pseudomonas aeruginosa*		1		1	2
*Salmonella* group D				1	1
*Streptococcus pneumoniae*	1				1
*Enterococcus faecium*		1			1
*Escherichia coli*				1	1
*Haemophilus influenzae*			1		1
Subtotal	1	3	9	8	21
Outcome					
ICU admission	0	0	4	1	5
Need tPA/DNase	0	0	0	1	1
Surgical intervention	0	0	1	0	1
Death	0	1	2	2	5

^a^ Including multiple anaerobes (*n* = 2), Escherichia coli and Corynebacterium stratum (*n* = 1), Staphylococcus aureus and Actinomyces spp (*n* = 1), S. aureus and Enterococcus faecium (*n* = 1), Campylobacter rectus and Parvimonas micra (*n* = 1).

**Table 3 microorganisms-11-00303-t003:** Outcomes of patients with pleural empyema in the pre- and post-COVID-19 periods.

Outcomes	Pre-COVID-19 Period (*n* = 1014)	Post-COVID-19 Period (*n* = 499)	% (95% CI) Difference
LOS, median ± IQR	24 ± 25 days	23 ± 21 days	
ICU admission	11.3%	13.6%	+2.3% (−1.1% to 6.0%)
IPFT	2.5%	8.4%	+5.9% (3.5% to 8.8%)
Surgical intervention	10.5%	7.8%	−2.7% (−5.6% to 0.5%)
Episode death	18.4%	20.6%	+2.2% (−1.9% to 6.6%)

IQR, interquartile range; IPFT, intrapleural fibrinolytic therapy.

**Table 4 microorganisms-11-00303-t004:** Potential risk factors for adverse patient outcomes in multivariate analysis.

	$\hat{\beta }\;$ Coefficient	Odds Ratio (95% Confidence Interval), Logistic Regression			
	Longer LOS	ICU Admission	IPFT	Surgical Management	Death
Age	1.36 (0.14 to 2.58)	1.00 (0.99–1.02)	1.01 (0.99–1.02)	1.00 (0.99–1.01)	1.03 (1.02–1.05) *
Male sex	5.71 (1.20 to 10.21)	0.97 (0.67–1.40)	1.51 (0.86–2.62)	1.42 (0.92–2.19)	1.21 (0.87–1.68)
CCI score	−0.48 (−1.36 to 0.40)	0.92 (0.85–1.00)	0.95 (0.84–1.07)	0.78 (0.70–0.87) *	1.16 (1.10–1.23) *
Etiology					
Polymicrobial	10.04 (3.96 to 16.11) *	1.66 (1.06–2.59)	0.57 (0.25–1.30)	1.54 (0.90–2.63)	1.66 (1.10–2.52)
Anaerobes ^a^	−13.08 (−18.22 to −7.94) *	1.13 (0.76–1.68)	0.32 (0.16–0.66) *	0.56 (0.34–0.91)	0.58 (0.40–0.85) *
*S. anginosus* ^a^	−15.07 (−20.17 to −9.96) *	1.16 (0.78–1.72)	0.26 (0.13–0.51) *	0.47 (0.28–0.79) *	0.43 (0.29–0.64) *
*S. aureus* ^a^	−2.40 (−8.76 to 3.96)	2.32 (1.20–4.49)	0.45 (0.18–1.12)	1.74 (1.04–2.90)	1.00 (0.66–1.51)
*S. pneumoniae* ^a^	−7.47 (−16.02 to 1.08)	1.52 (0.70–3.28)	0.64 (0.17–2.28)	0.24 (0.06–1.02)	1.50 (0.90–2.48)
Season (c/f autumn)					
Winter	−2.17 (−8.11 to 3.76)	1.05 (0.66–1.67)	0.85 (0.39–1.95)	0.92 (0.56–1.54)	1.05 (0.70–1.55)
Spring	−3.04 (−8.98 to 2.91)	1.10 (0.71–1.71)	1.03 (0.50–2.10)	1.00 (0.61–1.66)	0.94 (0.64–1.37)
Summer	1.28 *(−4.07 to 6.64)	1.24 (0.78–1.95)	1.45 (0.72–2.93)	1.16 (0.70–1.93)	0.71 (0.47–1.07)

IPFT, intrapleural fibrinolytic therapy. * *p* < 0.005. ^a^ Pleural empyema containing the organism.

## Data Availability

All relevant data have been included in the manuscript.

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
