# Peer review of "Changes in Etiology and Clinical Outcomes of Pleural empyema during the COVID-19 Pandemic"

_microorganisms, 2023, doi:10.3390/microorganisms11020303_

Round 1

Reviewer 1 Report

My comments

Main text

R1. When using a term for the first time, it is necessary to use full term and add the abbreviation next to it, instead of starting with an abbreviation

R2. Lines 61-65, in the introduction section are irrelevant to your topic, please remove that part

R3. I would like to know the impact of the Covid pandemic on the incidence of pleural empyema because given your results the prevalence of pleural empyema in the pre-Covid-19 period seems higher with 1014/1513 cases than in post Covid-19 with 499/1553 cases. Please add the plausible explanations

R4. In children aged 0-17 years, the average number of pleural empyema decreased from 18.4;4.8 cases per year in the pre Covid-19 period to 2.0;2.9 cases per year in the post covid-19 period. Please add the plausible explanations

Reviewer 2 Report

Comments:

1. In the Abstract section, the authors concluded that “We postulate that this is related to poor dental hygiene and delayed presentation of pneumonia to hospitals.”. But in this study, there are no results related to dental hygiene. So, in my opinion, there is no basis to support the conclusion.

2. In Figure 2, it would be better if the data pre-COVID-19 and post-COVID-19 of the microbial cause of pleural empyema according to age groups is all provided.

3. In Figure 1, several red wavy lines appear under the words. They should be removed.

4. For the tables, the standard format is a three-line table.

5. In Table 1, what is the meaning of “Totalb”? I guess that it should be “Total”.

6. In Table 4, there is a blank column.

7. It is apparent that there are two spaces between some words, for example, in “IPFT, intrapleural fibrinolytic  therapy” and “namely length  of stay”. Please check the whole manuscript carefully and revise them.

8. The format of REFERENCES is not consistent.

Reviewer 3 Report

The manuscript entitled " Changes in the etiology and clinical outcomes of pleural empyema during the COVID-19 pandemic" by Chan et. al. specifically compares the number and species of microbial pathogens in the pleural cavity of 1513 patients admitted in public hospitals in the city of Hong Kong. The seven-year data which the authors mined were divided into pre and post COVID period and analyzed accordingly. It was observed that among children, the average number of pleural empyema decreased from 18 cases per year in the pre-COVID-19 period to 2 cases per year in the post COVID-19 periods.  This situation was totally opposite in the case of adult patients. There is an abrupt increase in the proportion of polymicrobial pleural empyema in the post COVID-19 in adult patients. The author postulate that universal masking and school closure contributed considerably to decline in invasive disease during the pandemic era in the children. But the authors do not explain in a clear term why polymicrobial pleural empyema has increased drastically in the adult patients in the post COVID period. Although the data from 499 patients in the current study was post COVID-19 but the authors have not reported anywhere in the manuscript the status of COVID-19 in any of these patients. How many of these patients actually suffered from COVID-19 before being diagnosed for pleural empyema. That is the main drawback of the present study. It is possible that many of these patients might have caught COVID-19 and due to having a disturbed immunological status many of the reported invasive bacterial and other opportunistic pathogens have established themselves as secondary bacterial infection. The authors may need to provide a scientific explanation to the above points before this paper is published.

Round 2

Reviewer 2 Report

1. The name of the genus and species like pneumococcal pneumonia should be in italics.

Reviewer 3 Report

The authors have given a plausible explanation to my query and therefore I do not have any objection if the modified version of this manuscript is accepted for publication.
